# Clinical Simulation in Palliative Care for Undergraduate Nursing Students: A Randomized Clinical Trial and Complementary Qualitative Study

**DOI:** 10.3390/healthcare12040421

**Published:** 2024-02-06

**Authors:** Ana Alejandra Esteban-Burgos, Jesús Moya-Carramolino, Miriam Vinuesa-Box, Daniel Puente-Fernández, María Paz García-Caro, Rafael Montoya-Juárez, Manuel López-Morales

**Affiliations:** 1Department of Nursing, Faculty of Health Sciences, University of Jaén, 23071 Jaén, Spain; aesteban@ujaen.es; 2Gynecological-Obstetrics Nursing Specialist Residence, Vall d’Hebron Barcelona Hospital Campus, 08035 Barcelona, Spain; jesus.moya@vallhebron.cat; 3Clinical University Hospital San Cecilio, 18016 Granada, Spain; miriam.vinuesa.sspa@juntadeandalucia.es; 4Department of Nursing, Faculty of Health Sciences, University of Granada, 18016 Granada, Spain; mpazgc@ugr.es (M.P.G.-C.); rmontoya@ugr.es (R.M.-J.); malomorales@ugr.es (M.L.-M.); 5Instituto de Investigación Biosanitaria ibs.GRANADA, 18012 Granada, Spain; 6Mind, Brain and Behavior Research Institute, University of Granada, 18011 Granada, Spain; 7Primary Care Emergency Service, Andalusian Health System, 18013 Granada, Spain

**Keywords:** nursing students, nursing, palliative care, simulation training, clinical trial, self-efficacy, emotional intelligence

## Abstract

Background: a lack of adequate training in palliative care leads to a greater emotional burden on nurses. Purpose: to assess the effect of a simulation using standardized patients on self-efficacy in palliative care, ability to cope with death, and emotional intelligence among nursing students. Methods: a randomized clinical trial and qualitative study. A total of 264 nursing students in a palliative care module completed the Bugen, trait meta-mood, and self-efficacy in palliative care scales after active participation in the simulation (n = 51), watching the simulation (n = 113), and the control group (n = 100). An ANOVA with a multi-comparative analysis and McNemar’s tests for paired samples were calculated. Active participants were interviewed, and a thematic analysis was conducted. Results: there was an improvement after the assessment in all three groups assessed for coping with death (*p* < 0.01), emotional intelligence (*p* < 0.01), and self-efficacy (*p* < 0.01). In addition, the active group improved more than the observer group and the control group in coping with death, attention, and repair. The students in the interviews identified sadness and an emotional lack of control. Conclusions: the simulation improved nursing students’ self-efficacy in palliative care. This effect was partially stronger in the active group.

## 1. Introduction

Palliative care (PC) is an essential area of knowledge in the holistic approach to patient and family care [1]. It is defined as the active and holistic care of people of all ages with serious health-related suffering due to serious illness, and gaining special relevance and importance for patients nearing the end of life that aims to improve the quality of life of the patients, their families, and their caregivers [2].

Evidence suggests that providing adequate training in managing severe illness, communicating effectively, and providing psychosocial and spiritual support throughout nursing study can improve nursing professionals’ skills in end-of-life situations. This training results in improved attitudes towards PC among nurses [3], improved anxiety levels among nurses when caring for patients at the end of their life [4], and even improved decision-making regarding therapeutic alternatives among patients [5]. In contrast, a lack of specific training in PC leads to greater insecurity when caring for these patients in professional practice and higher burnout rates as a result of witnessing suffering and death first-hand [6].

Recent studies have reported that nursing students and new graduates lack skills and confidence when providing PC [7]. Limited clinical exposure to end-of-life care scenarios throughout the degree has become a concern in nursing education, as it can affect the caregiving skills of future nurses [8].

Simulations can help to reduce the gap from nursing theory to clinical practice [9]. It may be defined as an event or situation performed to mimic clinical practice as accurately as possible [10]. The benefits of clinical simulations include the acquisition of clinical knowledge and concepts, the understanding and application of cognitive, psychomotor, and communication skills, the promotion of clinical reasoning, and problem-solving without causing harm to real patients [11].

In a simulation, a standardized patient is a person trained to represent a patient in a realistic and repeatable way in all clinical situations or settings. Standardized patients interact with trainees in the simulation setting and can provide feedback on their performance [12]. This type of simulation is currently being used to train students in the health field and provides several benefits, including increased empathy and communication, critical thinking, and reflection skills, among others [13]. Simulation-based training is an excellent opportunity for nursing students to experience caring for patients in palliative and end-of-life situations, which can be challenging and stressful [14].

Simulation studies have so far focused on communication-related aspects [15,16], but recent studies have highlighted how simulation training can improve attitudes towards end-of-life care [17] and self-confidence related to end-of-life care [18]. The use of trained actors confers greater realism and fidelity to the scenarios presented in communication skill training programs. Recent studies [19,20] have shown that high-fidelity simulations with actors improved undergraduate nursing students’ communication skills and attitudes toward communication in complex situations involving chronicity and end-of-life care.

However, other variables, such as self-efficacy in PC, ability to cope with death, and emotional intelligence, have not been explored enough.

Firstly, self-efficacy is an individual’s assessment of their ability to organize and implement the courses of action required to achieve the designated objectives [21]. Individuals with a high level of professional self-efficacy set higher goals for themselves and persist when faced with difficulties, which they view as challenges rather than threats [22]. Secondly, coping with death has been defined as the skills and abilities that professionals possess to cope with death, as well as their attitudes and beliefs about their skills and abilities [23]. Finally, emotional intelligence is defined as a set of perceptions of our emotional world, how well we think we are at compressing, managing, and using both our own and others’ emotions [24]. Nurses caring for people at the end of their life need to show not only a high level of coping with death but also a high level of self-efficacy and emotional intelligence that enables them, among other things, to communicate effectively with patients and families and to prevent syndromes such as compassion fatigue.

Simulations require a high consumption of both human and technological resources, which could be a shortcoming in developing this methodology in low- and middle-income countries. One way to maximize the potential effect of a single simulation event is by using videos or streaming [25]. While a reduced number of students actively participate in a simulation scenario, a few students observe this interaction through a monitor or television. A recent study highlighted that active and observing learners show similar outcomes in knowledge and attitudes toward end-of-life care [26] after an intervention based on a simulation, but little was known about the effect of the simulation on self-efficacy in PC, coping with death, or emotional intelligence in both the groups. On the other hand, qualitative studies have evaluated simulation scenarios [27,28], although they have focused on the communication process or their general perception of the simulation.

This study aims to assess the effect of a clinical simulation with standardized patients on active and observing nursing students’ self-efficacy in PC, ability to cope with death, and emotional intelligence. A complementary objective is to explore the influence of sociodemographic variables (sex, age, and experience of death) on the variables. We established the hypothesis that training involving a simulation with standardized patients might increase self-efficacy in PC, ability to cope with death, and emotional intelligence compared to standard training. The effect might be higher in active learners than those solely observing the simulation. Finally, it describes how the students interpret their own emotions through the qualitative analysis of interviews conducted after they participate in the scenario.

Most published studies on simulations and PC are cross-sectional or cohort studies with no control group. [29]. In contrast, this is a randomized clinical trial, which provides a higher level of evidence to this research field.

## 2. Materials and Methods

### 2.1. Design

A randomized, non-blinded clinical trial (RCT) was conducted, assessing the effect of a simulation-based intervention using standardized patients on self-efficacy in PC, ability to cope with death, and emotional intelligence among second-year (out of a total of a four-year degree) undergraduate nursing students. A complementary qualitative study with a thematic analysis of semi-structured interviews was conducted, focused on the management of emotions of the students who participated in the scenarios.

### 2.2. Setting

The study was conducted at University of Granada as part of a compulsory palliative care 150 h module. The palliative care module is in the second year of a four-year undergraduate nursing program.

All students enrolled in the module during the 2018–2019, 2019–2020, and 2021–2022 academic years were invited to participate. The intervention and data collection procedure were not carried out in the 2020–2021 academic year due to the restrictions imposed on university teaching due to COVID-19. Every academic year, students are divided into three groups of 40 students. Two of the groups participated in the simulation scenarios, and the other was assigned to the control group. The sample size for the RCT was calculated using G*Power [30] for an expected effect size of 0.5 and α = 0.05.

A total of 264 from the 280-target student population participated in the study (dropout rate = 5.71%). The students who dropped out did not respond to the questionnaires (n = 16). The participants were divided into the active learning group (19.5%; n = 51), the observer learning group (42.8%; n = 113), and the control group (n = 100; 37.9%). The control and intervention groups were allocated according to their administrative teaching groups (A, B, and C). Blinding was not possible due to the nature of the intervention. In lieu of the intervention, the control group was taught what had been covered in previous academic years using audio-visual materials and focus groups. As is explained in the intervention section, the students who participated in the simulation scenarios and the qualitative study were randomly selected.

### 2.3. Intervention

Three scenarios relating to communication and emotional self-regulation in PC were enacted. The scenarios mirrored emotionally complex health and social situations and were developed through a consensus methodology by a team of PC faculty along with an undergraduate student and three nurses with postgraduate training in end-of-life care (Table 1).

The team that developed the scenarios provided the actors with a portfolio that included the objectives of the simulation, a thorough explanation of the situation to be developed, and the possible emotional responses of the learners and suggested responses to them. The actors had the opportunity to ask any questions they had about the simulation, as well as to offer suggestions about the interpretation [12].

The scenarios were implemented in small groups of 15–20 students. The classrooms used props to increase the environmental fidelity of the simulation scenario.

For each small group, two students were randomly selected using a lottery to take on the roles of nursing professionals (active learning). The order in which the students participated in each scenario was determined randomly, and the next student was unable to watch the previous student’s performance.

The students in the group who had not been selected watched the scenarios from the debriefing room on a video screen (observer learning). The duration of the scenarios ranged from 5 to 20 min.

After completing the scenarios, the randomly selected students re-joined the other students to discuss how the students managed the simulated situation [31]. After this teamwork, a debriefing session was held with the instructor. The debriefing aimed to explore students’ perceptions of the most important emotions in each scenario and to assess the performance of the students who had been selected to participate [32].

After the interaction, the actors were invited to the debriefing session so that the students could ask them questions and express their difficulties, and the actors could express their impressions about the attitude of the students.

The scenarios were part of the compulsory contents of the PC course. The PC teachers did not force anyone to actively participate. If the randomly selected student refused to participate, another student was selected, and neither refused the offer. The interaction with the standardized patients was not assessed for the course evaluation.

For a further description of the intervention, visit our webpage: https://paliativosugr.wordpress.com/

### 2.4. Instruments

To assess the effect of the intervention on the intervention group and the control group in the RCT, three questionnaires, which were translated and validated in Spanish, were administered to the two groups at the beginning and the end of the module.

-Bugen’s coping with death scale: Bugen [33] created this scale to operationalize perceived competence in the face of death. It contains 30 items rated on a Likert scale ranging from one (strongly disagree) to seven (strongly agree). It has a Cronbach’s alpha of 0.86 in this sample, which is similar to Spanish reliability data (α = 0.80) [34]. Based on the scale scores, coping with death is rated as inadequate (<105), adequate (105–157), or optimal (>157).-Trait meta-mood scale-24 (TMMS-24): The TMMS-24 is used for assessing emotional intelligence [35]. It consists of 24 items on the awareness and regulation of feelings, which are subdivided into three dimensions: emotional attention, emotional clarity, and emotional repair. These items are rated on a Likert scale ranging from one (strongly disagree) to five (strongly agree), with different cut-off points based on the sex of the respondent and scores classified as inadequate, adequate, or excellent. Higher scale scores indicate a greater ability to manage emotions, except for the emotional attention subscale, where high scores may indicate excessive attention to others’ emotions. The scale shows adequate reliability, with Cronbach’s alpha values of 0.89 for all subscales in this sample, which is similar to Spanish reliability data (α = 0.80) [36].-The self-efficacy in palliative care (SEPC) scale was developed in the United Kingdom. [37]. The reliability and validity of the Spanish version of the scale were determined using nurses and nursing students [38]. This study shows a Cronbach’s alpha value of 0.95 for the total scale, which is similar to the Spanish validation study (α = 0.94). The SEPC consists of 23 items assessing the perceived efficacy of communication, physical patient management, psychosocial/spiritual patient management, and multi-professional teamwork. Each behavior or skill is assessed using a 1–10 Likert scale ranging from “very anxious” to “very confident”. Higher scores indicate higher perceived efficacy among students or professionals on the overall scale and the different subscales.-An ad hoc form was used, including sociodemographic variables such as gender (men/women), age, previous training in health sciences (yes/no), and whether they had experienced a serious condition in their families or the death of a close family member (yes/no).

After the module, the questionnaires were administered again (post-test). The students were asked to identify their forms using the same alphanumeric code they had used in the pre-test to match both questionnaires (pre-test and post-test).

An ad-hoc semi-structured interview was conducted with the students who actively participated after each simulation scenario. The students responded to the interview when they finished the interaction with the standardized patient and before the debriefing session. The questions focused on students’ emotional management:How did you feel?Why do you think you feel like this?Do you think your emotions have influenced your responses?Do you think your performance could have helped in a real case? Do you think it would be easier or more difficult for you in a real case?

### 2.5. Analysis

For the RCT analysis, the continuous variables were described using means and standard deviations, and the discrete variables were described using frequency. The Chi-square test was used to check that there were differences in sample characteristics between the groups (gender, previous training in health sciences, religion, and experiences of serious conditions in their families or the death of a close family member). A unidirectional ANOVA with a Bonferroni post-hoc analysis was used to check that there were no significant differences in the study variables between the control group and the intervention groups at the start of the intervention (Bugen, TMMS, and SEPC). ANOVA-repeated measure tests were performed to verify the efficacy of the intervention on the studied parameters. The Bonferroni ad hoc test was used to see the differences between the groups (active learning, observer learning, and control).

McNemar’s change test was used to explore the differences in the percentages of the students who exhibited an excellent ability to cope with death (Bugen scale) and adequate emotional self-regulation (TMMS) before and after the intervention. The values for the TMMS subscales were grouped into adequate/excellent and inadequate. In the case of the variable TMMS attention, inadequate attention was considered as either too little or too much emotional attention.

All the tests were parametric, as the study variables were normally distributed as per the Kolgomorov-Smirnov test. The statistical significance threshold for all the tests was set at *p* < 0.05. The data were analyzed using IBM’s SPSS© (version 22) software.

A thematic analysis was performed for the qualitative analysis according to the work proposed by Joffe & Yardley [39]. This kind of analysis involves familiarization with the data, the generation of initial codes, and the searching, reviewing, and naming of themes according to the patterns, similarities, and differences among the codes. According to the aim of the study, we considered the questions of the semi-structured interviews as potential themes. The coding and selection of themes were reviewed and agreed upon by two independent researchers. Atlas.ti software (version 7.0) was used for all the analyses.

## 3. Results

### 3.1. Sample Description

A total of 264 students participated in the RCT, 83% of whom were female (n = 219). The mean age was 20.94 years old (SD = 3.962). A total of 119 (45.4%) participants reported professing a religion. Additionally, 26.2% (n = 69) had previous professional training in health sciences. Only 8% (n = 21) had experienced an accident or serious condition, while 75% (n = 198) had experienced a serious condition of a family member, and 79.2% (n = 209) had experienced the death of a close family member. No statistical differences were observed among the different groups regarding sample characteristics (Appendix A).

A total of 54 students, of whom 72% were women, participated in the qualitative study. The most common age among the participants was 19 years.

### 3.2. Pre-Test Results

The participants obtained a mean score of 117.84 (SD = 23.157) on Bugen’s scale. According to the Bugen scale, 33.3% of the students (n = 88) displayed inadequate coping, 62.5% displayed adequate coping (n = 165), and only 4.2% displayed optimal coping in the pre-test (n = 11). The students scored the highest on the TMMS dimension attention scale (*M* = 29.746; SD = 6.018) and the lowest on clarity (*M* = 25.704; SD = 5.765). According to the scale itself, 44.4% of the men (n = 20) and 38.4% of the women (n = 84) paid adequate attention to emotions; 51.1% of men (n = 23) and 52.6% of women (n = 122) obtained adequate values for emotional clarity, and 71.1% of men (n = 32) and 59.4% of women (n = 130) showed adequate repair. Regarding the SEPC, the dimension in which the students were the least self-confident at the start of the module was psychosocial management (*M* = 5.015; SD = 2.228).

There were no significant differences in the initial scale scores based on gender, religion, previous health science training, or whether the students had experienced a serious condition or the death of a close family member. No significant differences were found between the intervention and control groups at the start of the intervention based on the studied variables.

### 3.3. Effect of the Intervention

All students, regardless of whether they were in the control group or the intervention group, showed an increase in the variables analyzed after the PC module (Table 2).

After the PC module, the percentage of students who had an optimal ability to cope with death as per Bugen’s scale increased from 4.1% (n = 4.1) to 39.3% (n = 86) (*p* < 0.001).

Concerning the TMMS-24 scale, the proportion of women with an adequate score on the clarity subscale increased from 61.6% to 79.4% (*p* > 0.001), and the percentage of women with adequate mood repair increased from 72.6% to 81.3% (*p* = 0.004) (Table 3).

When calculating the differential effect of the intervention on the three experimental groups, significant differences were identified in the intervention groups in the coping with death and emotional regulation scales. The increase in the total Bugen’s coping with death score was higher in the active group compared to the observer (*p* = 0.006) and control group (*p* = 0.003). On the other hand, the TMMS attention score was higher in the active learning group than in the observer learning group (*p* = 0.009). The TMMS repair score increase was higher in the active learning group compared to the observer learning group (*p* = 0.006) and the control group (*p* = 0.013).

According to the McNemar test, the percentage of students whose ability to cope with death was optimal was somewhat higher in the active learning group (58.0%; n = 29) than in the observer learning group (34.6%; n = 37) and control group (33.3%; n = 32), although the differences were significant in both cases (*p* < 0.001).

Regarding the TMMS-24 scores, the differences in the percentages could only be calculated for women, as the number of men was too small. An increase in the percentage of women with adequate/excellent clarity was observed in both the observer (*p* = 0.008) and control groups (*p* = 0.009).

### 3.4. Qualitative Study

Nerves, insecurity, and sadness were the most prevalent feelings in the students. They did not know what they could do to help or how to react to some of the standardized patients’ attitudes. Some students attributed their feelings to the fact that it was the first time they faced a scenario similar to the one they would find in their professional lives.

“*I felt sadness for her situation and for the feelings and emotions she has experienced, and also anguish for not knowing how to steer the situation.*”(20-2 M19)

“*I have felt both pity for their situation and helplessness for not knowing how to act*”(22-13 F19)

“*Many emotions have arisen in me, such as fear of not doing things right, (..) I had never faced anything like this, so I have also felt concern and even anxiety for not acting properly or not knowing how to react.*”(20-1 F19)

Although the students highlighted that it was a positive feeling of empathy, they expressed fear of harming the other person and did not know how far they should let themselves be carried away by their own feelings. Some students felt that they could not control their own emotions and felt blocked.

“*I did not believe I had the authority to tell him how to feel, and I was afraid of hurting him.*”(20-3 F20)

“*I was stuck and did not know what to say or how to help the patient. I was more focused on what to say when the patient finished speaking than really listening to the patient*”(19-2 F19)

“*To be honest, I have felt pretty powerless. I felt bad for not knowing exactly what to tell him to make him feel better.*”(22-5 F19)

Although the participants reported that in the scenarios, they had the certainty that they were not harming the patient or relative, the students believed that it would be easier for them to act in a real situation because they would not feel judged or observed. The students highlight the plausibility of the scenarios.

“*In the scenario, I had the tranquility of knowing that it was not real and that I could not harm the patient with my words.*”(20-10 M20)

“*Being in front of a camera has influenced my way of acting or my nervousness.*”(20-2 F19)

“*I have learned that these types of cases are not ideal. People can blame you and take out their anger and fear against the staff.*”(22-12 M19)

“*I liked it very much because these situations worried me and seemed difficult to me, and I liked being able to simulate it and get closer to that reality. This practice is very useful*”(22-15 F21)

## 4. Discussion

The aim of the study was to assess the effect of an intervention based on simulated scenarios involving standardized patients and actresses on self-efficacy in PC, ability to cope with death, and emotional intelligence among nursing students and describe how students manage their emotions in the scenarios. Significant differences in all the variables were identified in the three groups (active learning, observer learning, and control) after completing the PC module. The active learning group showed a higher increase in coping with death, emotional attention, and emotional repair than the observer learning and control groups.

Echoing our results, the scores on the communication subscale of the SEPC were higher in men than in women in the validation study for the Spanish version of this scale [38]. However, the results described in the scientific literature regarding variables relating to emotional intelligence and sex are mixed. Like our study, Alconero-Camarero et al. (2018) [40] reported no significant differences between men and women when assessing the impact of a high-fidelity dummy intervention using the TMMS-24 scale. In another study on learning skills involving Spanish university students using the same tool [41], women exhibited greater emotional attention than men. Further studies are needed to explore the influence of the gender variable on emotions and end-of-life care.

No significant differences were identified in the variables analyzed between the students who professed a religion and those who did not. Studies suggest that professing a religion may improve attitudes towards end-of-life care [42]. In any case, religious rituals and professionals’ behavior towards dying individuals vary from culture to culture, which should be taken into account when extrapolating these results to other settings.

Regarding death-related experiences, a high proportion of participants in our study reported having had a sick family member or having experienced the death of a close family member. Our figures differ from other studies. Dimoula et al. (2019) [43] asked students if they had had any end-of-life experiences involving a close family member, finding that 42.2% had such an experience. On the other hand, students in a study by Kirkpatrick et al. (2017) [29] were asked if they had had one, two, or three end-of-life experiences; the percentage of those who had had at least one experience (93.2%) was even higher than the proportion in our study. However, there is no consensus in the scientific literature about the role of previous end-of-life experiences in end-of-life care. Studies have pointed out that previous experiences do not seem to play a key role [29,43], whereas other studies highlighted that those who had experienced the death of relatives or friends had more positive attitudes toward caring for the dying [44,45]. Further studies should explore how previous experiences regarding end-of-life might affect PC learning.

The results of this study suggest that there has been an improvement in the ability to cope with death as measured with the Bugen scale in both the control group and the intervention groups. It is well known that PC learning increases death competence [46] and attitudes toward the care of dying patients [17,43] for nursing students. Nevertheless, in our results, Bugen’s total score was higher in the active group compared to the observer and control group. It can be concluded that participating actively in simulation experiences could offer an additional improvement in coping with death. Although no studies were found that evaluate coping with death, our results partially contradict the results reported by Kirkpatrick, Cantrell, & Smeltzer (2020) [26], where a small group of student observers showed a similar betterment in attitudes toward caring for the dying than active participants after a simulation experience. Further qualitative or mixed-methods studies should explore how simulations affect coping with death in both groups (active and observers).

Regarding self-efficacy in PC, both groups improved their scores on communication, patient management, and teamwork compared to their initial scores. Our results agree with those reported by Tamaki et al. (2019) [18], who assessed the effect of a simulation with standardized patients and showed significant improvements in knowledge, physical assessment skills, psychological care skills, and self-confidence in PC.

Regarding emotional intelligence, nursing students’ scores were in the high range on all subscales, confirming previous studies [47]. This could explain the moderate effect of the training received on these variables among the students in both the intervention and control groups. Although no ceiling effect has been previously identified for the TMMS scale, more attention is needed on measuring emotional intelligence in trained populations such as nursing students.

ANOVA-repeated measures revealed that the increase in the TMMS repair score was higher in the active learning group than in the observer and control groups after completing the simulation scenarios. This is an important result because high reparation scores are linked to more favorable attitudes towards patients at the end of their life and their families [48], less death anxiety, and higher levels of self-esteem [49]. Although some studies have reported TMMS subscale improvement after simulations [40,50], this is, to our knowledge, the first study to compare active and observer learners with a control group. More attention needs to be paid to emotional intelligence variables in further studies to corroborate our results.

Regarding the results from the qualitative point of view, Mainey et al. [51] reflected how students who are exposed to clinical-simulated scenarios, like those in our study, feel nervous, insecure, and uncomfortable, as it was the first time they faced these circumstances and felt observed and judged. Despite this, after the experience of facing these scenarios, the students indicated that they felt more confident about dealing with this kind of situation in the future. This finding is in line with the greater effect of the intervention on self-efficacy in end-of-life care shown in the RCT.

Nagore-Ancona & Rodríguez (2017) [52] pointed out, like our study, the ambivalence of the emotions felt by the students, highlighting how some students are not able to control emotions, so this might affect the possibility of providing adequate care to patients. This finding may partially explain the low rates of emotional repair shown in the RCT.

Kirkpatrick, Cantrell, & Smeltzer (2017) [29] concluded in their review that the use of professional actors in simulations has a positive effect on students’ knowledge and self-efficacy in PC. The students also reported more engagement and satisfaction with the simulation with the use of live actors than with high-fidelity manikins.

Nurses and nursing students dealing with end-of-life patients and their families might significantly benefit from developing coping-with-death mechanisms, self-efficacy in palliative care, and emotional intelligence. These skills not only promote nurses’ well-being but also improve the quality of care provided to patients and their families during this sensitive time. Simulation-based learning utilizing actors as standardized patients might contribute to developing these variables because it allows nurses and nursing students to encounter and manage emotionally challenging and high-fidelity scenarios in a controlled setting, as well as post-simulation debriefing sessions, which encourage reflection and coping strategies.

Our study has limitations that should be considered. All the variables were evaluated for the three scenarios, so it is not possible to determine the relative impact of each simulation on learning outcomes. In the RCT, blinding was not possible because of the academic and administrative organization of the institution. Furthermore, the small number of men in our sample prevented us from performing some analyses on emotional self-regulation, as the TMMS was scored differently depending on gender. However, due to the gender difference in the sample, further studies with a male sample are needed to confirm these results. It should be noted that during the academic year of 2019–2020, university teaching had to be virtualized due to the COVID-19 pandemic. This did not affect the intervention carried out with the simulation scenarios but could affect the impact of the CP course. Concerning the qualitative study, results cannot be extrapolated to other contexts due to the low number of participants, but they can help to interpret data shown in the RCT.

## 5. Conclusions

In PC learning, the use of a simulation using actors as standardized patients increases coping with death, emotional attention, and emotional repair. This improvement is higher in students who actively participated in simulation scenarios, and, to a lesser extent, in the observer learning group and the control group.

However, the qualitative data show that students felt nerves, insecurity, and sadness and that they did not know how far they should be carried away by their feelings or have felt blocked by not controlling their own emotions.

Simulation with actors as standardized patients shows great benefits for PC learning, but more attention needs to be paid to the role of students’ emotional intelligence.

## Figures and Tables

**Table 1 healthcare-12-00421-t001:** Description of the intervention.

		Intervention Simulation Scenario				Control
	Name of the Practice	Role of the Actor/Actress	Simulation Setting	Prevailing Emotions	Description	Practice in the Control Group
2nd week	Mood identification and intervention	Terminally ill patient	Patient’s home	Denial/anger	She refuses to acknowledge her situation and becomes aggressive when it is pointed out to her.	Excerpts from the film *One True Thing* (Universal Pictures, 1998) and a class discussion.
3rd week	Emotional intelligence	Healthcare worker (male nurse)	Storage room in a hospital ward	Sadness/guilt	He feels guilty about the death of a patient.	Clinical cases on paper and class discussion.
6th week	Communicating with advanced chronic patients and their families	Family caregiver	Primary care nursing practice	Concern/anxiety	He or she takes a collusion of silence approach.	Ad hoc video on how to ease the collusion of silence and a class discussion.

**Table 2 healthcare-12-00421-t002:** Effect of the intervention on the three groups (ANOVA).

			Initial		Final		ANOVA IntraGroup *p*	ANOVA Intergroup *p*	Post hoc
	N		M	SD	M	SD			
Bugen	A	51	123.82	21.06	158.21	21.10	<0.001	=0.002	A > O *p* = 0.006 A > C *p* = 0.003
	O	113	116.86	25.72	146.48	23.60			
	C	100	115.92	20.71	145.67	23.51			
TMMS attention	A	51	31.27	4.93	32.71	5.46	<0.001	=0.012	A > C (*p* = 0.009)
	O	113	29.67	5.86	31.06	5.41			
	C	100	29.05	6.59	30.13	5.59			
TMMS clarity	A	51	26.71	5.83	30.10	5.98	<0.001	0.121	
	O	113	25.56	5.45	28.71	5.94			
	C	100	25.36	6.07	28.01	6.18			
TMMS repair	A	51	28.71	5.91	32.02	5.84	<0.001	0.005	A > O (*p* = 0.006) A > C (*p* = 0.013)
	O	113	26.95	6.63	28.78	5.89			
	C	100	26.95	6.36	29.09	6.03			
SEPC communication	A	51	5.75	1.68	7.70	1.20	<0.001	0.128	
	O	113	5.11	1.88	7.53	1.39			
	C	100	5.44	1.88	7.35	1.37			
SEPC physical manag.	A	51	5.13	1.67	7.62	1.24	<0.001	0.374	
	O	113	5.02	2.03	7.52	1.17			
	C	100	4.94	2.15	7.25	1.39			
SEPC Psychosocial manag.	A	51	5.35	1.73	7.75	1.37	<0.001	0.134	
	O	113	5.03	2.34	7.58	1.33			
	C	100	4.83	2.33	7.36	1.43			
SEPC teamwork	A	51	6.58	2.06	8.42	0.90	<0.001	0.167	
	O	113	6.24	2.15	7.98	1.24			
	C	100	6.23	2.30	8.03	1.30			
SEPC total	A	51	5.82	1.29	7.91	0.91	<0.001	0.102	
	O	113	5.43	1.63	7.67	1.08			
	C	100	5.49	1.72	7.54	1.12			

A: active. O: observer. C: control.

**Table 3 healthcare-12-00421-t003:** Effect of the intervention on TMMS-24 scores according to the total and exposure groups in females only (McNemar’s test for related samples).

			Initial		Final		
			n	%	n	%	*p*
TTMS Attention **	Active n = 41	Inadequate	11	26.8	18	43.9	=0.092
		Adequate	30	73.2	23	56.1	
	Observer n = 98	Inadequate	41	41.8	34	34.7	=0.311
		Adequate	57	58.2	64	65.3	
	Control n = 80	Inadequate	32	40.0	26	32.5	=0.418
		Adequate	48	60.0	54	67.5	
TMMS clarity	Active n = 31	Inadequate	14	34.1	8	19.5	=0.146
		Adequate	27	65.9	33	80.5	
	Observer n = 98	Inadequate	38	38.8	21	21.4	=0.008 *
		Adequate	60	61.2	77	78.6	
	Control n = 90	Inadequate	32	40.0	16	20.0	=0.009 *
		Adequate	48	60.0	64	80.0	
TMMS repair	Active n = 31	Inadequate	10	24.4	4	9.8	=0.227
		Adequate	31	75.6	35	85.4	
	Observer n = 98	Inadequate	28	28.6	17	17.3	=0.108
		Adequate	70	71.4	76	77.6	
	Control n = 90	Inadequate	22	27.5	13	16.3	=0.093
		Adequate	58	72.5	67	83.8	

* *p* ≤ 0.01. ** For the initial TMMS attention and final TMMS attention variables, the attention was considered inadequate if the students showed too little or too much emotional attention.

## Data Availability

The data presented in this study are available upon request from the corresponding author.

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
