# Peer review of "Clinical Simulation in Palliative Care for Undergraduate Nursing Students: A Randomized Clinical Trial and Complementary Qualitative Study"

_healthcare, 2024, doi:10.3390/healthcare12040421_

Round 1

Reviewer 1 Report

Comments and Suggestions for Authors

I would like to express my gratitude for the opportunity to review this manuscript, which focuses on a relevant and emerging theme: the need for training for healthcare professionals (or future professionals) to improve care and the performance of these professionals when caring for patients with palliative needs. The text is presented clearly and logically, with sections making sense in relation to each other. The study design is suitable for the proposed objectives, and its analysis is appropriate.

However, I have some questions that seem to require reformulation or improvement.

  1. The title does not reflect the purpose of the manuscript. Typically, this kind of formulation suggests an impact on the provision of palliative care to others. This work is equally commendable and relevant, but I believe the title should reflect the goal of the study. e.g.: Clinical simulation using actors in palliative care teaching "in nurse students' performance"...

  2. Line 41: It is important not to mix two concepts that are different, even if they touch on each other. Palliative care should not be especially for people at the end of life, even though it is essential in this phase. It is, in fact, an important challenge for palliative care to expand beyond end-of-life patients, starting in earlier stages. I suggest that instead of "especially of those nearing the end of life," the sentence be formulated as "gaining special relevance and importance for patients nearing the end of life."

  3. Lines 82-84: It seems more appropriate for the discussion section, especially because the intervention used does not involve video or streaming methodology.

  4. Line 121: It is important to mention the sample size obtained. I have doubts about whether any dropout rate was considered or not, and if not, why this decision was made.

  5. Lines 123-125: There is no information on how participants were allocated to each study arm. Later in the text, the way elements for active training were selected in each group is understood, but it is relevant to understand here how students were allocated.

  6. Some information is missing about how the intervention, specifically the applied scenarios, was developed. Expert consensus? Another methodology? This information is crucial because without it, the methodology cannot be replicated by another group. I accessed the link on the page, but there is something missing about the program/scenario development process.

  7. In the methodology section, it is necessary to clarify the variables collected, in addition to the results of the instruments used. Some of the data presented in the sample description - which is relevant - lacks previous context. Were more relevant data collected that have not been presented? What was the reason for this choice? I believe this methodology section can be improved to convey information more clearly.

  8. Table 2, the last line p 0.00 should be replaced by p<0.01.

  9. The qualitative study needs to be developed. It is described in the methodology section that a thematic analysis was performed, but these data are not presented in the qualitative analysis section. What exists is a reflection with some quotes. Although relevant, it might be an overstatement to define this section as a structured qualitative analysis. I suggest rethinking and reformulating if the authors intend to keep this analysis, which, I emphasize, seems relevant.

  10. Lines 239-254: A very relevant piece of information is missing here. Not only should the values of the scores between groups be presented, but also the characteristics of the students in each arm. This point is closely related to the previously mentioned lack of allocation strategy. This data should be presented in a table that, although not included in the manuscript, should be in supplementary material. Without this, it is unclear whether the initial characteristics of the students may also have explanatory power, weakening the results and conclusions.

  11. Line 410: "Furthermore, the small num- 403 bers of men in our sample." This information is relevant but depends a lot on the men/women ratio among the students in the evaluated year. We do not have data to understand whether this information has an impact on representativeness.

Throughout the text, there are some typos, mostly related to missing punctuation near bibliographic references.

Best regards.

Comments on the Quality of English Language

The text is clear, easy to understand, and without major typos. I identified only a few punctuation issues.

Author Response

Reviewer 1

Reviewer (R): I would like to express my gratitude for the opportunity to review this manuscript, which focuses on a relevant and emerging theme: the need for training for healthcare professionals (or future professionals) to improve care and the performance of these professionals when caring for patients with palliative needs. The text is presented clearly and logically, with sections making sense in relation to each other. The study design is suitable for the proposed objectives, and its analysis is appropriate. However, I have some questions that seem to require reformulation or improvement.

Authors (A): Thank you very much for your commentaries and suggestions.  We hope that you will find the responses satisfactory.

(R): The title does not reflect the purpose of the manuscript. Typically, this kind of formulation suggests an impact on the provision of palliative care to others. This work is equally commendable and relevant, but I believe the title should reflect the goal of the study. e.g.: Clinical simulation using actors in palliative care teaching "in nurse students' performance"...

(A):  According to your suggestion we have changed the manuscript´s title.

Line 2: Clinical simulation in palliative care for undergraduate nursing students:  a randomised clinical trial and complementary qualitative study.

(R): Line 41: It is important not to mix two concepts that are different, even if they touch on each other. Palliative care should not be especially for people at the end of life, even though it is essential in this phase. It is, in fact, an important challenge for palliative care to expand beyond end-of-life patients, starting in earlier stages. I suggest that instead of "especially of those nearing the end of life," the sentence be formulated as "gaining special relevance and importance for patients nearing the end of life."

(A): The sentence has been modified according to your recommendation

Line 41: It is defined as the active and holistic care of people of all ages with serious health-related suffering due to serious illness, and gaining special relevance and importance for patients nearing the end of life, that aims to improve the quality of life of the patients, their families and their caregivers[2]

(R): Lines 82-84: It seems more appropriate for the discussion section, especially because the intervention used does not involve video or streaming methodology.

(A): Thanks for your suggestion. We believe that this phrase in this part of the introduction section is necessary to highlight the importance of including active learning and an observer learning group, who watched the scenarios from the debriefing room on a video screen (streaming).

(R): Line 121: It is important to mention the sample size obtained. I have doubts about whether any dropout rate was considered or not, and if not, why this decision was made.

(A): Thanks for the suggestions. The dropout rate was added to the methods section.

Line 135: A total of 264 from 280 target student population participated in the study (Dropout rate=5,71%). The students who dropped out did not respond to the questionnaires (n=16).

(R): Lines 123-125: There is no information on how participants were allocated to each study arm. Later in the text, the way elements for active training were selected in each group is understood, but it is relevant to understand here how students were allocated.

(A): The control and intervention groups were allocated according to administrative teaching groups (A, B  and C). As it is explained later on, the students who participated in the active learning group were randomly selected from the intervention group. This has been clarified in the methods section:

Line 138: The control and intervention groups were allocated according to administrative teaching groups (A, B and C).

(R): Some information is missing about how the intervention, specifically the applied scenarios, was developed. Expert consensus? Another methodology? This information is crucial because without it, the methodology cannot be replicated by another group. I accessed the link on the page, but there is something missing about the program/scenario development process.

(A): Thanks for your commentary. The scenarios mirrored emotionally complex health and social situations and were developed through a consensus methodology by a team of PC faculty along with an undergraduate student and three nurses with postgraduate training in end-of-life care.

This has been clarified in the methods section:

Line 147: The scenarios mirrored emotionally complex health and social situations and were developed through a consensus methodology by a team of PC faculty along with an undergraduate student and three nurses with postgraduate training in end-of-life care. (Table 1).

(R): In the methodology section, it is necessary to clarify the variables collected, in addition to the results of the instruments used. Some of the data presented in the sample description - which is relevant - lacks previous context. Were more relevant data collected that have not been presented? What was the reason for this choice? I believe this methodology section can be improved to convey information more clearly.

(A): An ad hoc form was included that allowed us to collect relevant variables such as gender, age, previous training in health sciences (yes/no), and whether they had experienced a serious condition in their families or the death of a close family member (yes/no). No other variables were collected. The values of the variables have been added to the methods section.

Line 210: An ad hoc form including sociodemographic variables such as gender (men/women), age, previous training in health sciences (yes/no), and whether they had experienced a serious condition in their families or the death of a close family member (yes/no).

In the results section we have highlighted that there were no significant differences in the collected variables (Line 261).

(R): Table 2, the last line p 0.00 should be replaced by p<0.01.

(A): Thank you for your comment. The typo has been corrected.

(R): The qualitative study needs to be developed. It is described in the methodology section that a thematic analysis was performed, but these data are not presented in the qualitative analysis section. What exists is a reflection with some quotes. Although relevant, it might be an overstatement to define this section as a structured qualitative analysis. I suggest rethinking and reformulating if the authors intend to keep this analysis, which, I emphasize, seems relevant.

(A): A thematic analysis was performed for qualitative analysis, according to the work proposed by Joffe & Yardley. This kind of analysis involves the familiarization with data, the generation of initial codes, and the searching, reviewing, and naming of themes, according to the patterns, similarities, and differences among codes. According to the aim of the study, we have considered the questions of the semi-structured interview as potential themes.

This has been clarified in the methods section.

Line 248: This kind of analysis involves the familiarization with data, the generation of initial codes, and the searching, reviewing, and naming of themes, according to the patterns, similarities, and differences among codes. According to the aim of the study, we have considered the questions of the semi-structured interview as potential themes.

New quotations have been added to the results section.

Line 321: “I have felt both pity for their situation and helplessness for not knowing how to act” (22-13 F19)

Line 347: “I liked it very much because these situations worried me and seemed difficult to me, and I liked being able to simulate it and get closer to that reality. This practice is very useful” (22-15 F21)

(R): Lines 239-254: A very relevant piece of information is missing here. Not only should the values of the scores between groups be presented, but also the characteristics of the students in each arm. This point is closely related to the previously mentioned lack of allocation strategy. This data should be presented in a table that, although not included in the manuscript, should be in supplementary material. Without this, it is unclear whether the initial characteristics of the students may also have explanatory power, weakening the results and conclusions.

(A): Thanks for the suggestion. A Table has been added as supplementary material to show the characteristics of the different groups. This has been clarified in the methods section.

Line 261: No statistical differences were observed among the different groups regarding sample characteristics [Appendix A]

(R): Line 410: "Furthermore, the small numbers of men in our sample." This information is relevant but depends a lot on the men/women ratio among the students in the evaluated year. We do not have data to understand whether this information has an impact on representativeness.

(A): Thanks for the suggestion. Unfortunately, we cannot provide the men/women ratio of the courses that have been evaluated. Previous literature has shown that the proportion of men is similar in other samples of undergraduate nursing students. However, a phrase has been added to the limitations section:

Line 447: However, due to the gender difference in the sample, further studies with a male sample are needed to confirm these results.

(R): Throughout the text, there are some typos, mostly related to missing punctuation near bibliographic references.

(A): The text has been revised and typos have been corrected.

Reviewer 2 Report

Comments and Suggestions for Authors

First of all, I want to congratulate the authors for submitting a manuscript with a lot of quality and with a very pertinent theme.

O manuscrito tem o título "Clinical simulation using actors in palliative care teaching: a randomized clinical trial and complementary qualitative study” e tem como objetivo “assess the effect of a simulation using standardized patients on self-efficacy in palliative care, ability to cope with death, and emotional intelligence among nursing students”. It is a randomized trial with mix-method characteristics, although it is not presented as a mix-method study.

The introduction effectively sets the context for the study. However, it would be beneficial to more clearly articulate the unique contribution of this study compared to existing literature, especially focusing on the use of actors in clinical simulation.

The literature review is comprehensive could be enhanced by discussing more recent studies, especially those focusing on simulation in palliative care education.

The study design is well-articulated. It would be helpful to provide more detail about the selection and training of actors used in the simulations. How were they prepared to ensure consistency and realism in their performances?

Clarification on the blinding process within the randomized trial could strengthen the methodology section.

The results are presented clearly. It might be useful to include more detailed statistical analysis to provide a deeper understanding of the findings.

Including more direct quotes from the qualitative part could enhance the understanding of students' experiences.

The discussion ties the results back to the literature review effectively. It would be beneficial to discuss the implications of these findings for future research and practical applications in nursing education.

Consider discussing the limitations of the study in more depth, such as potential biases or the generalizability of the findings.

The conclusion summarizes the findings well. Expanding on the practical implications of the research and suggesting areas for future study could make this section more impactful.

The conclusion summarizes the findings well. Expanding on the practical implications of the research and suggesting areas for future study could make this section more impactful.

Author Response

Reviewer 2

Reviewer (R): First of all, I want to congratulate the authors for submitting a manuscript with a lot of quality and with a very pertinent theme.

Authors (A): Thank you very much for your kind words. We hope you will find the changes we have made satisfactory.

(R): O manuscrito tem o título "Clinical simulation using actors in palliative care teaching: a randomized clinical trial and complementary qualitative study” e tem como objetivo “assess the effect of a simulation using standardized patients on self-efficacy in palliative care, ability to cope with death, and emotional intelligence among nursing students”. It is a randomized trial with mix-method characteristics, although it is not presented as a mix-method study.

(A): A mixed-methods study refers to a research approach that combines both quantitative and qualitative methods within a single investigation or study to gain a more comprehensive understanding of a phenomenon or research question. Although synergies between quantitative and qualitative data are highlighted in the discussion section, there is not a specific integration of data. That is the reason why we believe that this study cannot be understood as a mixed-methods study.

 (R): The introduction effectively sets the context for the study. However, it would be beneficial to more clearly articulate the unique contribution of this study compared to existing literature, especially focusing on the use of actors in clinical simulation. The literature review is comprehensive could be enhanced by discussing more recent studies, especially those focusing on simulation in palliative care education.

(A): Thanks for the suggestion. A paragraph has been added to highlight the unique contribution of this study.

Line 89: Nurses caring for people at the end of life need to show not only a high level of coping with death but also a high level of self-efficacy and emotional intelligence that enables them, among other things, to communicate effectively with patients and families and to prevent syndromes such as compassion fatigue.

New text has been added to the introduction section discussing recent studies.

Line 68: Simulation-based training is an excellent opportunity for nursing students to experience caring for patients in palliative and end-of-life situations, which could be challenging and stressful [14].

Line 73: The use of trained actors confers greater realism and fidelity to the scenarios presented in communication skills training programs. Recent studies [19,20]  have shown that high-fidelity simulation with actors improved undergraduate nursing students’ communication skills and attitudes towards communication in complex situations involving chronicity and end-of-life care.

Two references have been included:

Abad-Corpa, E., Guillén-Ríos, J. F., Pastor-Bravo, M. D. M., & Jiménez-Ruiz, I. (2023). Assessment of high fidelity simulation with actors in palliative care in nursing students: a mixed methods study. Enfermeria clinica (English Edition), 33(6), 401–411. https://doi.org/10.1016/j.enfcle.2023.10.003  

Hamdoune, M., & Gantare, A. (2021). Teaching palliative care skills via simulation-based learning. International journal of palliative nursing, 27(7), 368–374. https://doi.org/10.12968/ijpn.2021.27.7.368

Escribano, S., Cabañero-Martínez, M. J., Fernández-Alcántara, M., García-Sanjuán, S., Montoya-Juárez, R., & Juliá-Sanchis, R. (2021). Efficacy of a Standardised Patient Simulation Programme for Chronicity and End-of-Life Care Training in Undergraduate Nursing Students. International journal of environmental research and public health, 18(21), 11673. https://doi.org/10.3390/ijerph182111673

(R): The study design is well-articulated. It would be helpful to provide more detail about the selection and training of actors used in the simulations. How were they prepared to ensure consistency and realism in their performances?

(A): Thanks for the suggestion. Further explanations have been added to the methods section.

Line 152: The team that developed the scenarios provided the actors with a portfolio that included the objectives of the simulation, a thorough explanation of the situation to be developed, and the possible emotional responses of the learners and suggested responses to them. The actors had the opportunity to ask any questions they had about the simulation, as well as to make suggestions about the interpretation.

(R): Clarification on the blinding process within the randomized trial could strengthen the methodology section.

(A): Blinding was not possible due to the nature of the intervention. This has been clarified in the methods section.

Line 139: Blinding was not possible due to the nature of the intervention.

It has also been highlighted in limitations (Line 444)

 (R): The results are presented clearly. It might be useful to include more detailed statistical analysis to provide a deeper understanding of the findings.

(A): New analyses have been added to the methods section, according to another reviewer’s suggestion.

Line 229: The chi-Square test was used to check that there were differences in sample characteristics between groups (gender, previous training in health sciences, and experiences of serious condition in their families or the death of a close family member).

(R): Including more direct quotes from the qualitative part could enhance the understanding of students' experiences.

(A): Thanks for the suggestion. New quotations have been added to the results section.

Line 321: “I have felt both pity for their situation and helplessness for not knowing how to act” (22-13 F19)

Line 347: “I liked it very much because these situations worried me and seemed difficult to me, and I liked being able to simulate it and get closer to that reality. This practice is very useful” (22-15 F21)

 (R): The discussion ties the results back to the literature review effectively. It would be beneficial to discuss the implications of these findings for future research and practical applications in nursing education.

(A): Thanks for your suggestion. A new paragraph has been added to the discussion section.

Line 433: Nurses and nursing students dealing with end-of-life patients and their families might benefit significantly from developing coping with death mechanisms, self-efficacy in palliative care, and emotional intelligence. These skills not only promote the nurses' well-being but also improve the quality of care provided to patients and their families during this sensitive phase. Simulation-based learning utilizing actors as standardized patients might contribute to developing these variables because allows nurses and nursing students to encounter and manage emotionally challenging and high-fidelity scenarios in a controlled setting, as well as post-simulation debriefing sessions encourage reflection and coping strategies.

(R): Consider discussing the limitations of the study in more depth, such as potential biases or the generalizability of the findings.

(A): Thanks for your suggestion. Two limitations have been added to the discussion section, regarding gender differences and COVID-19 restrictions for the 2019-2020 academic year.

Line 445: Furthermore, the small number of men in our sample prevented us from performing some analyses on emotional self-regulation, as the TMMS was scored differently depending on gender. However, due to the gender difference in the sample, further studies with a male sample are needed to confirm these results.

It should be noted that during the academic year 2019-2020, university teaching had to be virtualized due to the COVID-19 pandemic. This did not affect the intervention carried out with the simulation scenarios but could affect the impact of the CP course.

(R): The conclusion summarizes the findings well. Expanding on the practical implications of the research and suggesting areas for future study could make this section more impactful.

(A): Thanks for your suggestion. A new paragraph has been added to the discussion section.

Line 433: Nurses and nursing students dealing with end-of-life patients and their families might benefit significantly from developing coping with death mechanisms, self-efficacy in palliative care, and emotional intelligence. These skills not only promote the nurses' well-being but also improve the quality of care provided to patients and their families during this sensitive phase. Simulation-based learning utilizing actors as standardized patients might contribute to developing these variables because allows nurses and nursing students to encounter and manage emotionally challenging and high-fidelity scenarios in a controlled setting, as well as post-simulation debriefing sessions encourage reflection and coping strategies.

Reviewer 3 Report

Comments and Suggestions for Authors

There are concerns regarding the study design. The randomization process is unclear as the clinical simulation module is mandatory for all students in the course. 

Therefore, it is necessary to provide justification for this point and explain the method of randomisation.

Additionally, the sample of students was collected from three different academic years, but there is a significant variable that sets them apart: the COVID-19 pandemic. During the 2019-2020 academic year, all learning activities were adapted to online methodologies. The following year, face-to-face activities were extremely limited and also adapted to online methods.  It is important to note that these circumstances significantly impacted the teaching activity, which differs from what is explained in the paper.

How did you determine the size of the control group? Why is it 37.9%? The control group has to be similar to the intervention group, and it isn't. 

Another important consideration is that the control group must consist of students in the same circumstances as those participating in the intervention. Therefore, they cannot be students from a previous year, who may not have the same level of knowledge and who were assessed at a different point in time. This point is not clearly explained in the paper. 

The sampling procedure for the qualitative analysis needs to be clarified, including the final sample and justification for its selection. It should be noted that the study design does not cover qualitative analysis, but rather a mixed methodology approach.  

Additionally, the usual 'constant comparison' between different actors is not carried out in the qualitative study. The group of students has been treated as a single entity, despite the fact that the control group is expected to have entirely different circumstances.

Author Response

Reviewer 3

Reviewer (R): There are concerns regarding the study design. The randomization process is unclear as the clinical simulation module is mandatory for all students in the course. Therefore, it is necessary to provide justification for this point and explain the method of randomisation.

(A):  Thanks for the suggestion. New text has been added to the methods section to provide further explanations of the randomization method.

Line 135: A total of 264 from 280 target student population participated in the study (Dropout rate=5,71%). The students who dropped out did not respond to the questionnaires (n=16). Participants were divided into the active learning group (19.5%; n=51) observer learning group (42.8%; n=113) and control group (n=100; 37.9%). The control and intervention groups were allocated according to administrative teaching groups (A, B and C). Blinding was not possible due to the nature of the intervention.

(R):  Additionally, the sample of students was collected from three different academic years, but there is a significant variable that sets them apart: the COVID-19 pandemic. During the 2019-2020 academic year, all learning activities were adapted to online methodologies. The following year, face-to-face activities were extremely limited and also adapted to online methods.  It is important to note that these circumstances significantly impacted the teaching activity, which differs from what is explained in the paper.

(A): In the 2019-2020 academic year, the simulation scenarios took place before the lockdown and virtualization of university teaching was decreed in Spain. Nevertheless, we assume that there might have been an effect on the impact of the palliative care course in that academic year, which has been noted as a limitation in the discussion.

Line 449: It should be noted that during the academic year 2019-2020, university teaching had to be virtualized due to the COVID-19 pandemic. This did not affect the intervention carried out with the simulation scenarios but could affect the impact of the CP course.

In the academic year 2020-2021, due to the uncertainty caused by the restriction of teaching activities, it was decided not to carry out the simulation scenarios, as reflected in the methodology. A sentence has been added to the methods section to highlight this decision.

Line 129: The intervention and data collection procedure were not carried out in the 2020-2021 academic year due to the restrictions imposed on university teaching by COVID-19.

(R):  How did you determine the size of the control group? Why is it 37.9%? The control group has to be similar to the intervention group, and it isn't.

Thanks for your commentary. The sample size for the RCT was calculated using G*Power for an expected effect size of 0.5 and α =.05. As it has been explained in the methods section, participants were divided into the active learning group (19.5%; n=51) observer learning group (42.8%; n=113) and control group (n=100; 37.9%). The control and intervention groups were allocated according to administrative teaching groups (A, B and C).

A phrase has been added to clarify this process.

Line 138: The control and intervention groups were allocated according to administrative teaching groups (A B and C).

In an RCT, it is ideal to strive for balance between the number of participants in the intervention and the control group. However, it's not an absolute requirement. What's crucial is ensuring that both groups are comparable in terms of key characteristics that might affect the outcome being measured. For this reason, a Table has been added as supplementary material to show the characteristics of the different groups.

(R):  Another important consideration is that the control group must consist of students in the same circumstances as those participating in the intervention. Therefore, they cannot be students from a previous year, who may not have the same level of knowledge and who were assessed at a different point in time. This point is not clearly explained in the paper.

(A): Thanks for your commentary. As has been explained in the methods section (Line 128), the study was conducted as part of a compulsory Palliative Care module in the second year of a four-year undergraduate nursing programme, during three academic years. The content of the module was the same in all three academic years and was taught by the same lecturers, therefore, the level of knowledge and the assessment periods were similar.

(R):  The sampling procedure for the qualitative analysis needs to be clarified, including the final sample and justification for its selection. It should be noted that the study design does not cover qualitative analysis, but rather a mixed methodology approach. 

(A): Thanks. The sampling procedure for qualitative analysis has been clarified in the methods section (Setting).

Line 142: As is explained in the intervention section, the students who participated in simulation scenarios and the qualitative study were randomly selected.

Further details are provided later in the methods section (Intervention). (Line 152)

A mixed-methods study refers to a research approach that combines both quantitative and qualitative methods within a single investigation or study to gain a more comprehensive understanding of a phenomenon or research question. Although synergies between quantitative and qualitative data are highlighted in the discussion section, there is not a specific integration of data. That is the reason why we believe that this study cannot be understood as a mixed-methods study.

(R):  Additionally, the usual 'constant comparison' between different actors is not carried out in the qualitative study. The group of students has been treated as a single entity, despite the fact that the control group is expected to have entirely different circumstances.

(A): As has been explained in the previous commentary, only the students who participated in simulation scenarios answered the qualitative study interview, so it is expected that had comparable experiences.

Round 2

Reviewer 1 Report

Comments and Suggestions for Authors

The article has been revised in accordance with the comments, and I believe it is ready to be published.

Reviewer 3 Report

Comments and Suggestions for Authors

All indications of changes made to the authors have been carried out. The manuscript has been significantly improved. It is almost a transformation into another article.  In any case, I believe that the problematic points have been solved.